# A Literature Review on System Dynamics Modeling for Sustainable Management of Water Supply and Demand

Khawar Naeem [1] , Adel Zghibi [1,2,*] , Adel Elomri [1] , Annamaria Mazzoni [3] and Chefi Triki [4]

1 College of Science and Engineering, Hamad Bin Khalifa University, Qatar Foundation, Doha P.O. Box 34110, Qatar
2 Laboratory of Geological Resources and Environment, Department of Geology, Faculty of Sciences of Tunis, University of Tunis El Manar, Tunis 2092, Tunisia
3 Qatar Environment and Energy Research Institute, Hamad Bin Khalifa University, Qatar Foundation, Doha P.O. Box 34110, Qatar
4 Kent Business School, University of Kent, Canterbury CT2 7UL, UK
* Correspondence: azghibi@hbku.edu.qa or adel.zghibi@fst.utm.tn; Tel.: +974-70651948

**Abstract:** Water supply and demand management (WSDM) is essential for developing sustainable cities and societies. WSDM is only effective when tackled from the perspective of a holistic system understanding that considers social, environmental, hydrological, and economic (SEHEc) sub-systems. System dynamics modeling (SDM) is recommended by water resource researchers as it models the biophysical and socio-economic systems simultaneously. This study presents a comprehensive literature review of SDM applications in sustainable WSDM. The reviewed articles were methodologically analyzed considering SEHEc sub-systems and the type of modeling approach used. This study revealed that problem conceptualization using the causal loop diagram (CLD) was performed in only 58% of the studies. Moreover, 70% of the reviewed articles used the stock flow diagram (SFD) to perform a quantitative system analysis. Furthermore, stakeholder engagement plays a significant role in understanding the core issues and divergent views and needs of users, but it was incorporated by only 36% of the studies. Although climate change significantly affects water management strategies, only 51% of the reviewed articles considered it. Although the scenario analysis is supported by simulation models, they further require the optimization models to yield optimal key parameter values. One noticeable finding is that only 12% of the articles used quantitative models to complement SDM for the decision-making process. The models included agent-based modeling (ABM), Bayesian networking (BN), analytical hierarchy approach (AHP), and simulation optimization multi-objective optimization (MOO). The solution approaches included the genetic algorithm (GA), particle swarm optimization (PSO), and the non-dominated sorting genetic algorithm (NSGA-II). The key findings for the sustainable development of water resources included the per capita water reduction, water conservation through public awareness campaigns, the use of treated wastewater, the adoption of efficient irrigation practices including drip irrigation, the cultivation of low-water-consuming crops in water-stressed regions, and regulations to control the overexploitation of groundwater. In conclusion, it is established that SDM is an effective tool for devising strategies that enable sustainable water supply and demand management.

**Keywords:** system dynamics modeling (SDM); water supply and demand management (WSDM); groundwater sustainability; optimization; sustainable development



## 1. Introduction

Water supply and demand management (WSDM) is the backbone of enabling the sustainable development of any society. An effective WSDM comprises devising strategic policies that ensure sufficient water to meet socioeconomic and environmental demands. By 2030, it is forecasted that around 2 billion people will be affected by the water scarcity problem [1]. Approximately 45% of the global population will live in water-stressed regions

by 2050, and around 250 million people will lack access to clean water [2]. This makes water resource planning and management a fundamental pillar for societal sustainability [3]. The increasing socio-economic development of a growing population and commercial activities leads to a higher water demand [4]. The impact of climate change exacerbates this problem further, especially in arid and semi-arid regions where groundwater (GW) is overexploited due to poor long-term water management strategies [5]. This leads to the problem of GW depletion, seawater intrusion, and the degradation of the ecosystem as well as increased competition among different users within the water sector [6]. A holistic system under-standing is needed to devise effective strategies for water planning and management [7]. Furthermore, stakeholder engagement in water management is essential for planning and implementing strategies with little or no societal resistance [8,9]. Understanding and conceptualizing a holistic water resource system is a complex challenge, as it involves inter-action among the various SEHEc sub-systems [10,11]. This interaction among sub-systems is non-linear and dynamic and cannot be framed and modeled using linear approximations or mechanistic techniques, including multi-criteria decision models. Linear models lack the required framework to tackle the complexity of studying natural systems, including water resource management. These models also need to support the integration and interaction of sub-systems [12]. Hence, a comprehensive understanding of these systems requires a methodology that integrates the necessary sub-systems, models each system's complex feedback, identifies the root cause of the problem, and helps policymakers devise effective water planning and management strategies [13]. A holistic system approach that adopts all of the above-mentioned characteristics to model such a complex system is system dynamics modeling (SDM) [10].

SDM is a holistic systems approach which, using simulation software, can integrate the hydrological sub-system of a region and its socio-economic and environmental sub-systems to study the long-term impact of water resource policies from the region's water-governing institutes. SDM is a decision support system (DSS) that helps policymakers to understand such complex systems, design effective long-term water planning and management strategies, and capture the interacting sub-systems' dynamic feedback using the features of the stock and flow diagram (SFD) [14,15].

The broader applications of SDM in water resource management have been presented by several researchers. Chen and Wei [16] reviewed SDM applications in disaster man-agement, water resource security, and environmental resource management. The generic modeling applications of SDM were also reviewed by Gohari et al. [17], and they were categorized into descriptive SDM, predictive SD simulation models, and participatory SDM. Zomorodian et al. [18] extended the research of Gohari et al. [17] and surveyed several case studies applying SDM to solve water resource problems.

To date, an in-depth analysis of sustainable WSDM using the SDM framework remains unaddressed in the literature. This review article aims to fill this gap. In particular, this study discusses the findings of published articles and case studies that used the SDM approach as the prime tool to understand and analyze the complex WSDM and to achieve water resource sustainability. The key research questions investigated in this review study are the following:

Question 1: Is system dynamics modeling (SDM) an effective decision support tool for water supply and demand management?
Question 2: Does SDM foster sustainable development?
Question 3: How can system dynamics help policymakers to foster groundwater conservation?
Question 4: What are the different quantitative models that are used in conjunction with SDM to improve water resource management?

Following this introduction, the paper is structured as follows. Section 2 provides an overview of SDM, while the research methodology is presented in Section 3. A detailed content analysis on sustainable water supply and demand management is presented in Section 4. A summary of the key findings and research gaps is presented in Section 5. Finally, the conclusion and future recommendations are presented.

## 2. System Dynamics Modeling (SDM): An Overview

SDM is an approach that considers and integrates interacting sub-systems as a whole system [19]. SDM is used to identify the long-term behavior of an entire system given the change in the state of its sub-systems over a specific period [10]. SDM was first conceived by Forrester [20], and it was initially applied to the system analysis of a commercial business. In the last two decades, SDM has been actively used to study and analyze complex environmental and water resource management policies [18].

In water resource management, SDM captures the collective non-linear interaction effects and dynamics within the hydrologic, social, economic, and environmental sub-systems [21]. The generic review strategy adapted in this literature review is presented in Figure 1. In this way, a holistic understanding of the water resource system was developed [22].

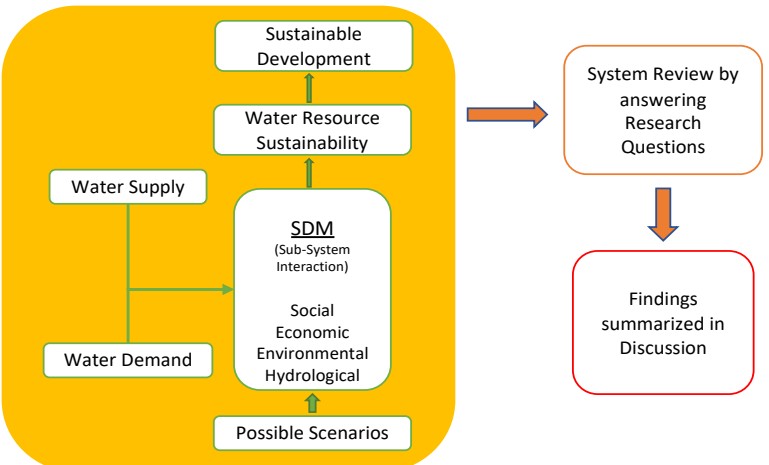

**Figure 1.** SDM application in water resource sustainability review strategy.

SDM has the potential for both qualitative and quantitative system analysis. The causal loop diagram (CLD) is a qualitative system analysis tool that helps to conceptualize and frame the overall system. CLD is actively used in water resource management studies [23].

The quantitative tool of SDM is called the stock flow diagram (SFD). In an SFD quantitative analysis, the system's current state is known as the stock. In a water resource system, the "*stock*" represents the water quantity (resource) available to serve the system, while the "*flow*" represents the "rate" at which the water (resource stock) is used [24]. It is shown mathematically in Equation (1):

$$Stock\ (t) = Stock\ (ts) + \int_{ts}^{t} (Inflow(s) - Outflow(s))\ ds \tag{1}$$

where the current time is represented by (*t*), while the initial time is represented by (*ts*). At any given time (*s*), the *inflow* and *outflow* are represented by *Inflow* (*s*) and *Outflow* (*s*), respectively [18].

SDM helps to identify system trends and determine the root causes of the system imbalance or perturbation. In this way, the system analyst uses the SDM model as a decision support system (DSS).

## 3. Methodology

This literature review analysis was conducted using two main academic databases: Google Scholar and Scopus. The keywords used for the search included "system dynamics*" OR "system dynamics modeling" AND "Water supply and demand" OR "Groundwater*" OR "Groundwater management". To match all words surrounding the keyword "System dynamics", the wildcard "*" was utilized. The time limit constraint for the literature review was up to August 2022. Interestingly, however, no research articles with the above-

stated keywords were found before 2002, which is in line with the findings of Zomorodian et al. [18], as shown in Figure 2.

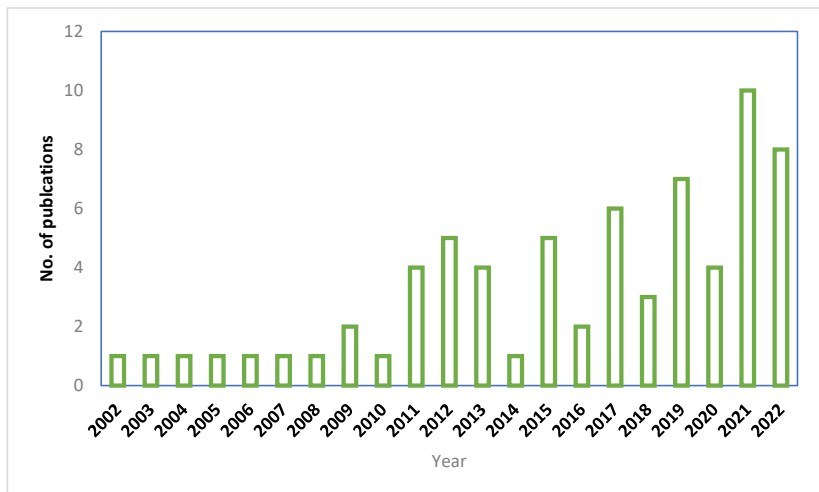

**Figure 2.** Articles using the studied keywords published per year.

This shows that researchers' interest in the application of SDM in WSDM and GW resource management began in late 2000. According to this methodology, the final number of articles selected for this review was 69, as shown in Figure 3. To match all words surrounding the keyword "System dynamics," the wildcard "*" was utilized. The relatively small number of articles can be attributed to the fact that the application of SDM to water resource management usually involves a cross-disciplinary team of researchers.

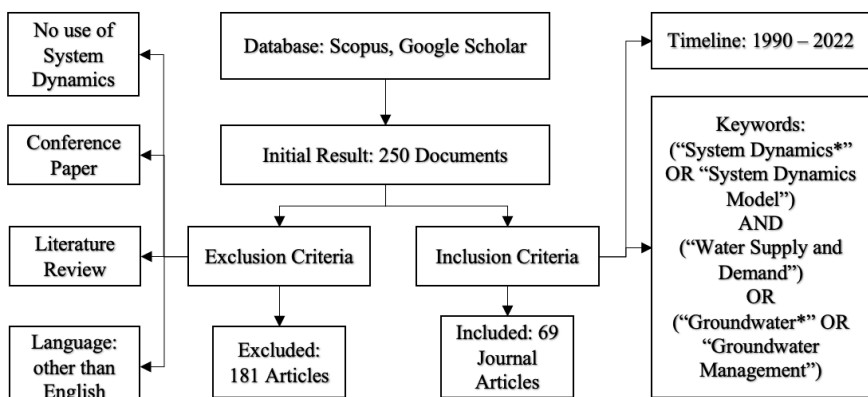

**Figure 3.** Flowchart of the review methodology.

The articles were published in 33 distinct journals, reflecting the interdisciplinary research interest of the scientific community in water resource management, as shown in Figure 4.

The potential relationship between the different research areas and challenges with the SDM approach were explored using a semantic network, as shown in Figure 5. This approach was developed using the open-source Biblioshiny version 4.1.2 software. SDM is a primary approach adopted by researchers to foster sustainable development by addressing water resource management issues. Therefore, SDM is used for water conservation and for balancing water supply and demand through the analysis of possible management scenarios, facilitating stakeholder participation and developing pragmatic solutions. Figure 5 shows the direction of the detailed content analysis in the subsequent section. The main issues related to water management are presented depending on the different research areas in which SDM was applied.

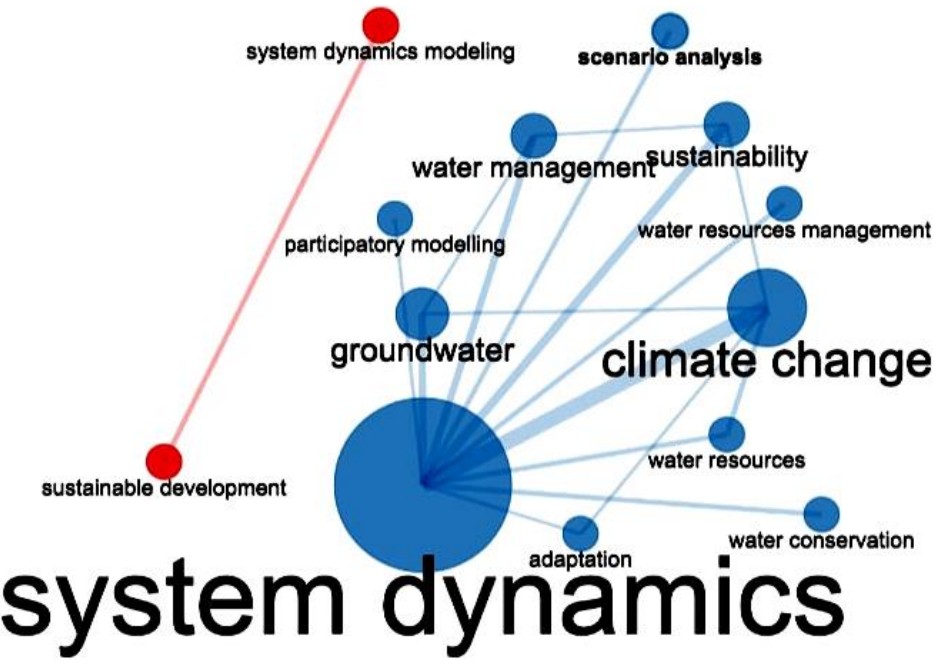

**Figure 4.** Relevant journals about the studied theme.

**Figure 5.** Semantic network relating SDM to water resource management research topics.

## 4. Results

The research areas included water scarcity problems, sustainable water supply management, demand management, sustainable development, climate change impact on water resources, and participatory modeling for effective policymaking, as shown in Table 1. The application of SDM in the identified research areas is explored in the subsequent section.

Sustainable WSDM is essential for the long-term sustainable development of any region. It is significant for policymakers to consider it while planning long-term resources and development projects. Failure to plan will have significant social, economic, and environmental repercussions.

WSDM has been studied by researchers using the SDM approach, and the focus has been on analyzing the impact of policies and strategies on the system's future state. Under the umbrella of this central research theme, the sub-categories of this research (or research themes) include the impact of GW sustainability, stakeholder engagement, sustainable development, climate change, irrigation, the reuse of treated wastewater, and the water–energy–food (WEF) nexus.

Hybrid models which combine SDM with other techniques were also explored and used to solve water planning and management challenges. The reviewed studies were further analyzed regarding their consideration of the social, environmental, hydrological, and economic (SEHEc) sub-systems, the use of CLD and SFD tools, model validation, the simulation software used, and the impact of climate change. This information is tabulated at the end of each section.

### 4.1. Sustainable Development Using WSDM

Sustainable development in any region is impossible without effective water planning and management [25]. Parallel to socioeconomic growth, water planning and management is pivotal to maintain the hydrological and environmental balance of the ecosystem.

The water resource management (WRM) of the Zayndh region (Iran) was analyzed by Madani and Mariño [15], with a focus on surface water and GW preservation. The complex system, with interacting socio-economic, environmental, and hydrological systems, was simulated in their study. Using the SDM approach, the researchers identified the flaws in the regional water policy and recommended a set of actions to decision makers. These actions included controlling the GW withdrawal per capita at a safe abstraction level, monitoring the per capita water consumption rate, considering the trans-basin water diversion, and increasing the water storage capacity to avoid water shortages. Similarly, Kotir et al. [26] developed a decision support system (DSS) using the SDM approach to analyze an African region's water resource management practices. Three different management policy scenarios were simulated. The impact of long-term policies was analyzed using the devised DSS. It was recommended that investment in water infrastructure projects is a better option compared to the expansion of agricultural land for the future sustainability of water resources.

To address the water supply and demand gap, the water supply and demand management system of an arid city in Iran was studied by Zarghami and Akbariyeh [27]. The water supply included GW, freshwater, and treated wastewater. A holistic study of the water distribution system was conducted using the SDM approach. The long-term impact of five different water policies was studied to mitigate the water shortage problem. The authors recommended conducting inter-basin water transfers and improving wastewater treatment efficiency. This would reduce water shortages by approximately 40% in the coming 10 years.

The water scarcity issue may hamper regional sustainable development. Shenzhen, China, has faced these challenges due to the expansion of the industries in the region. To address the challenge, the city's water distribution system was simulated for fifteen years using the SDM approach by Li et al. [28], and the water supply and demand ratio (WSDR) was analyzed. In the base case scenario, the WSDR was 0.9, which is inadequate. Analyzing the SDM scenario for the future 15 years and considering optimized parameter

values for the wastewater usage rate and the per capita domestic water demand, the WSDR improved to 1.03. Moreover, to achieve sustainable regional development, the estimation of water demand is essential. The estimation of the water demand helps policymakers to invest in water infrastructure projects at the right time to sustain regional socio-economic development [29], as was shown by Kamal et al. [30], who used Qatar as a case study and employed SDM. Based on historical data, the future energy and water demand for the next thirty years were estimated using a linear regression model. These estimates, as well as the limiting GW policy of 50 million $m^3$ and the usage of desalinated water of up to 50%, will result in an increase in the energy demand and $CO_2$ emissions. Similarly, Qi and Chang [31] used SDM to perform water demand forecasting for Florida (USA). GW over-exploitation is one of the main problems due to inefficient water management and planning [32], especially in arid and semi-arid regions, as discussed in the subsequent section.

The key findings of this section are tabulated in Table 1. The symbol (✓) means that the mentioned feature is considered in the research study.

**Table 1.** Reviewed articles on SD applications in sustainable development, considering the WSDM. H: hydrological modeling; S: social modeling; Ec: economic modeling; En: environmental modeling.

| Reviewed Studies | Location | Modeling Approach | Sub-System Modeling | | | | SDM (System Dynamic Modeling) | | | Software Used | Climate Change Impact |
|---|---|---|---|---|---|---|---|---|---|---|---|
| | | | H | S | Ec | En | CLD | SFD | validation | Vensim (V) or Stella (S) or Other (O) | |
| Madani and Mariño [15] | Iran | Policy analysis for effective water supply and demand management using SDM. | ✓ | ✓ | ✓ | ✓ | ✓ | ✓ | ✓ | S | ✓ |
| Wang et al. [26] | Ghana | Sustainable water resource management considering socio-economic and bio-physical sub-systems using SDM. | ✓ | ✓ | ✓ | | ✓ | ✓ | | S | ✓ |
| Zarghami and Akbariyeh [27] | Iran | Urban water system analysis using SDM. | ✓ | ✓ | ✓ | ✓ | ✓ | ✓ | | S | ✓ |
| Kotir et al. [28] | China | Water supply demand management using SDM. | ✓ | ✓ | ✓ | | ✓ | ✓ | ✓ | V | |
| Kamal et al. [30] | Qatar | Water efficiency impact on energy system using SDM. | ✓ | | ✓ | | ✓ | ✓ | ✓ | V | |
| Qi and Chang [31] | USA | Water demand estimation and management using SDM. | ✓ | ✓ | ✓ | | | ✓ | ✓ | | |

### 4.2. Groundwater (GW) Sustainability

WSDM is essential to avoid the overexploitation of GW, which occurs when the average recharge rate is lower than the average abstraction rate. To understand GW sustainability holistically, SDM is frequently used [33,34]. The overexploitation of GW has serious consequences regarding water quality degradation, a reduced natural GW level, land subsidence, and environmental and ecosystem degradation [35].

Several articles in our review address this challenge. The water scarcity issue in Tunisia was analyzed using SDM by Sušnik et al. [36]. The primary stressor causing water demand was the per capita water demand and the water demand of coastal cities. The concept of smart GW governance (SGG) was introduced by Barati et al. [37]. The Iranian GW system was analyzed using SDM. Using the novel concept of SGG, water policies were assigned a smartness index. The indices included GW sustainability, equitability, use efficiency, and democracy. Based on the SGG methodology, the GW governance in Iran needs serious policy interventions for resource sustainability.

The sustainability of GW resources in the arid region of the Nesa Plain, Iran, was investigated by Sardo and Jalalkamali [38]. It was noted that the GW level would be reduced by eight meters in the coming fifteen years if the current water withdrawal rate remained

the same. Moreover, an artificial GW recharge was suggested for water sustainability. Similarly, the critical decline in the GW of the Fairuzabad agricultural plain in Iran was studied by Mokhtar and Aram [39] using SDM. The failure of government policies to regulate agricultural practices using GW wells was identified. A 2% reduction in demand per year per hectare can lead toward the sustainable balance of the GW and can be achieved in the next fifteen years. Bates et al. [40] aided policymakers in effective planning at the basin level using SDM in California.

Hydro-economic modeling using SDM can be used to address the water scarcity problem, as addressed by Assaf [41]. The water scarcity problem faced by the Aman-Zarq Basin (AZB), Jordan, is due to excessive, uncontrolled GW extraction. It was found, using SDM simulation, that under the status quo of the business-as-usual scenario, the fossil aquifer reserves will be depleted in the next two decades. Adopting the same approach, Mahdavinia and Mokhtar [42] recommended allocating a budget for water in Iran to educate the stakeholders on the importance of GW conservation and increase water tariffs. One of the negative consequences of excessively pumping GW beyond the safe abstraction limit is land subsidence. Using SDM, Huang et al. [43] analyzed land subsidence in Taiwan. Sealing unauthorized wells and controlling illegal GW extraction in the region was recommended to avoid any possible harmful effects, including fracture damage and negative environmental impacts.

Efficient irrigation practices are essential for mitigating the water scarcity issue [44], which was studied by Mahdavi et al. [45] in the Azarshah Aquifer (Iran), using SDM. Horticultural and agricultural irrigation efficiency was recommended to bring equilibrium to the water resource system. An SDM simulation model of the complex dynamic interaction of SW–GW in an irrigation land in China was developed by Khan et al. [46] to analyze several irrigation management strategies. The objective was to improve water use efficiency and avoid possible salination problems by simulating different irrigation water management scenarios. This helped policymakers analyze the irrigation system's dynamic behavior over time using a simulation model conducted previously using the physical hydrological modeling approach.

The water system and related socioeconomic indicators of the Zayanderud Endorheic Basin, Iran, were analyzed by Enteshari et al. [47], using SDM. It was recommended to reduce the irrigation area by 10%. Similarly, Mir et al. [48] used SDM to study the Sistan region of Iran. The regional river water supply from Helmand, Afghanistan, was disrupted due to the impact of climate change. Therefore, the Sistan region has moved to rely on fossil water reserves. The best manner to maximize water conservation is to irrigate low-water-consuming crops. The key findings of the section are tabulated in Table 2. The symbol (✓) means that the mentioned feature is considered in the research study.

For effective water resource management and GW conservation, the impact of climate change must be studied and incorporated into scientific models [49]. The previously stated studies articulated water management policies and possible scenarios well, but most failed to capture the impact of climate change. This is explored in detail in the subsequent sub-section.

### 4.3. The Impact of Climate Change

Using SDM, the impact of climate change on the water resource of Hashtgerd, Iran, was assessed by Mehrazar et al. [50]. Using eighteen AOGCM (GEOS-5 atmosphere–ocean general circulation model) climate models, it was predicted that the exacerbation of the hot and dry conditions in the region due to climate change would lead to water shortages, with the agriculture sector being the most affected sub-system. Improving irrigation efficiency and cultivating drought-tolerant crops was recommended to avoid future water scarcity problems. The developed SDM helped the water resource planning department to allocate and justify the needed funds for investment in adoption plans to mitigate the impact of climate change on water resources and avoid possible future water shortages.

**Table 2.** Articles reviewed on applications of SDM in GW sustainability. H: hydrological modeling; S: social modeling; Ec: economic modeling; En: environmental modeling.

| Reviewed Studies | Location | Modeling Approach | Sub-System Modeling | | | | SDM (System Dynamic Modeling) | | | Software Used | Climate Change Impact |
|---|---|---|---|---|---|---|---|---|---|---|---|
| | | | Hy | S | Ec | En | CLD | SFD | Validation | Vensim (V) or Stella (S) or Other (O) | |
| Sušnik et al. [36] | Tunisia | System dynamics modeling (SDM) used to study the water scarcity to aid policymakers in decision making through a scenario analysis. | ✓ | ✓ | ✓ | ✓ | ✓ | | | O (Simile) | ✓ |
| Barati et al. [37] | Iran | To develop a smart water governance model for long-term water sustainability using SDM. | ✓ | ✓ | ✓ | ✓ | ✓ | ✓ | ✓ | V | ✓ |
| Sardo and Jalalkamali [38] | Iran | The SDM approach was used to model the recharge scenarios, considering climate change projections. | ✓ | ✓ | ✓ | ✓ | ✓ | ✓ | ✓ | V | ✓ |
| Mokhtar and Aram [39] | Iran | GW management integrating socio-economic and biophysical aspects, using SDM. | ✓ | ✓ | ✓ | | ✓ | ✓ | ✓ | V | |
| Bates et al. [40] | USA | Sustainable GW management using the SDM approach. | ✓ | | | | ✓ | | | | |
| Assaf [41] | Jordan | GW resource management using a hydro-economic SDM model. | ✓ | | ✓ | | | ✓ | | S | ✓ |
| Mahdavinia and Mokhtar [42] | Iran | Integrated GW assessment using SDM, considering the water security indicators. | ✓ | ✓ | ✓ | | ✓ | ✓ | ✓ | V | |
| Huang et al. [43] | Taiwan | Land subsidence analysis resulting from GW overexploitation, using SDM. | ✓ | | ✓ | ✓ | | | ✓ | V | |
| Mahdavi et al. [45] | Iran | GW sustainability analysis using SDM. | ✓ | ✓ | ✓ | | ✓ | ✓ | ✓ | V | |
| Enteshari et al. [47] | Iran | Water management considering socio-economic sub-systems using SDM. | ✓ | ✓ | ✓ | | ✓ | ✓ | ✓ | O | ✓ |
| Mir et al. [48] | Iran | GW resource sustainability study using SDM. | ✓ | ✓ | ✓ | | ✓ | ✓ | ✓ | O | |

In light of the impact of climate change, water scarcity and water resource management were addressed by Chen et al. [51] in an arid region of China. The main drivers for GW exploitation were inefficient water reuse and increasing commercial demand. A reduction in the irrigation of the land was recommended to control the overexploitation of GW. To address a similar issue, a DSS-ESPA (decision support system—Eastern Plain Aquifer) was developed using SDM [52]. The DSS-ESPA parameters included aquifer recharge and discharge, climate change, and GW–surface water conservation percentages.

Balali et al. [53] used SDM to study the effect of climate change on the GW behavior of the Hamdan aquifer, Iran. It was recommended to increase the price of regional irrigation water and energy for GW pumping, as this would help control the overexploitation of GW. Using SDM analysis, the impact of climate change on water resources in Vietnam was explored by Phan et al. [54], who found that rising sea levels are one of the main impacts causing the decline in the upstream flow of freshwater. This poses a severe problem for future water security in the region. To address this challenge, nature-based solutions (NBSs) were analyzed by Gómez Martín et al. [55] using SDM. Participatory modeling was essential for effectively implementing the NBSs, considering regional social and economic sustainability.

The interaction and impact of socioeconomic factors and future climate conditions on the water resource system of Faro, Sweden, was studied by Nicolaidis et al. [56], who suggested that because of climate changes, GW levels will decline to a critically low level in the coming twenty years. This will negatively impact housing and tourism development, with an average reduction of 6% and 15%, respectively. The impact of the increasing temperature, population growth, and greenhouse gases (GHGs) on water quality and quantity was studied by Duran-Encalada et al. [57] in Reynosa, Mexico. It was noted that the increasing water demand, reducing supply, and increasing temperature, mainly due to GHGs, will lead the resources toward depletion and deterioration. It was recommended that policymakers incentivize the industrial sector toward water recycling and reuse practices and invest in urban water distribution infrastructures to minimize losses.

Considering the impact of climate change on Hamdan, Iran, the sustainability of GW resources was studied by Afruzi et al. [58] using SDM. Reductions in irrigation land and the irrigation of low-water-consuming crops were recommended as the best strategies to achieve GW sustainability. Using SDM, the impact of climate change on the GW in the Zarine River Basin was explored by Baghanam et al. [59]. Using artificial neural networks (ANNs), they predicted a decrease of eight percent in the precipitation level and an increase of $0.6\ ^\circ$C. Thus, the GW recharge rate will be poor, and AQR was suggested to mitigate this issue. Similarly, the water demand management policies in Las Vegas, USA, were analyzed by Dawadi and Ahmed [60] using SDM. The impact of climate change on the water supply from the Colorado River revealed a decline in water quantity, considering fifteen global climate models and two emission scenarios. Using SDM, it was predicted that if the current water usage practices continue, there would not be enough water by 2035 to fulfill the demand. Several water conservation policies were simulated to reduce the demand, including smart-water appliances and increased water tariffs. It was claimed that even if no population growth is assumed, the water demand would not be met in the future 15 years with the current water usage practices.

The key findings of the section are tabulated in Table 3. The symbol (✓) means that the mentioned feature is considered in the research study.

**Table 3.** Reviewed articles on SD applications incorporating the impact of climate change. H: hydrological modeling; S: social modeling; Ec: economic modeling; En: environmental modeling.

| Reviewed Studies | Location | Modeling Approach | Sub-System Modeling | | | | SDM (System Dynamic Modeling) | | | Software Used | Climate Change Impact |
|---|---|---|---|---|---|---|---|---|---|---|---|
| | | | H | S | Ec | En | CLD | SFD | Validation | Vensim (V) or Stella (S) or Other (O) | |
| Mehrazar et al. [50] | Iran | Climate change impact on GW-surface water conservation strategies, using SDM. | ✓ | ✓ | ✓ | ✓ | ✓ | ✓ | ✓ | V | ✓ |
| Chen et al. [51] | China | Water resource management using SDM, considering climate change impact (GW-focused). | ✓ | ✓ | ✓ | ✓ | ✓ | ✓ | | | ✓ |
| Ryu et al. [52] | Iran | Water resource management under uncertain supply scenarios due to the impact of climate change, using SDM. | ✓ | ✓ | | ✓ | | ✓ | | | ✓ |
| Phan et al. [54] | Vietnam | Assessment of water vulnerability under climate change conditions, using SDM. | ✓ | ✓ | ✓ | | ✓ | ✓ | ✓ | | ✓ |
| Gómez Martín et al. [55] | Spain | Dynamic behavior assessment of nature-based solutions towards drought alleviation under several socio-economic and climate scenarios, using SDM | ✓ | ✓ | ✓ | ✓ | | ✓ | ✓ | S | ✓ |
| Nicolaidis Lindqvist et al. [56] | Sweden | Water supply and demand assessment under climate variability considering, hydro-socio sub-systems and using SDM. | ✓ | ✓ | | ✓ | | | | S | ✓ |
| Duran-Encalada et al. [57] | Mexico | Water resource management under imapct of climate change, using SDM. | ✓ | ✓ | ✓ | ✓ | ✓ | ✓ | ✓ | | ✓ |
| Afruzi et al. [58] | Iran | GW management considering variable climate scenarios, using SDM | ✓ | ✓ | | ✓ | ✓ | ✓ | ✓ | V | ✓ |
| Baghanam et al. [59] | Iran | Water resource management under the impact of climate change, using SDM. | ✓ | | | ✓ | ✓ | ✓ | ✓ | V | ✓ |
| Dawadi and Ahmad [60] | USA | Water demand management considering climate change and population growth, using SDM | ✓ | ✓ | ✓ | ✓ | | ✓ | | S | ✓ |

### 4.4. Stakeholder Participation

Stakeholder participation (SP) is essential for long-term viable, sustainable solutions and an effective water management policy [61]. Considering SP, the water resource system of Rafsajan Plain, Iran, was studied by Moghaddasi et al. [62] The physical parameters of the system were derived using a GW management system (GMS) modeled in MODFLOW. It was concluded that the most effective and socially viable policy is to use an efficient water irrigation system in addition to water diversion, which will significantly improve the region's water resilience and conservation. Pluchinotta et al. [63] modeled the SP for GW sustainability in Apulia, Italy, using the interaction space (IS) methodology. The IS concept was merged with SDM. Future climate projections and the interaction of the socio-economic parameters of an arid region of Pakistan were investigated by Alizadeh et al. [64] to advise on effective climate change policies to ensure SP since the current scenario, an increase in water demand of 3% by 2030, is projected to place high stress on regional water resources.

The increasing population and drought in Kherbad, Iran, threaten water resources, as studied by Ghasemi et al. [65]. The stakeholders were engaged in the problem analysis using SDM, and it was noted that under the same population growth and without increasing water resources, the annual water demand will increase by one percent, while the water availability will decrease by 1.5%. The implementation of a water pricing mechanism and per capita water consumption measures were recommended. Stave [66] conducted a study for effective SP in water resource management through a workshop organized in Nevada, USA. Using SDM, the participants could see and understand the counterintuitive results, such as that the outdoor water consumption reduction was much more effective than the indoor water consumption reduction.

The key findings of the section are tabulated in Table 4. The symbol (✓) means that the mentioned feature is considered in the research study.

**Table 4.** Reviewed articles on SD applications incorporating stakeholder engagement. H: hydrological modeling; S: social modeling; Ec: economic modeling; En: environmental modeling.

| Reviewed Studies | Location | Modeling Approach | Sub-System Modeling | | | | SDM (System Dynamic Modeling) | | | Software Used | Climate Change Impact |
| --- | --- | --- | --- | --- | --- | --- | --- | --- | --- | --- | --- |
| | | | H | S | Ec | En | CLD | SFD | Validation | Vensim (V) or Stella (S) or Other (O) | |
| Moghaddasi et al. [62] | Iran | The GW management policy analysis using SDM, based on the stakeholder framework. Scenarios are ranked based on Electoral College Voting. | ✓ | ✓ | ✓ | ✓ | | ✓ | | | ✓ |
| Pluchinotta et al. [63] | Italy | Considering multi-stakeholder decision making, analyzed effective irrigation water management practices using SDM. | | ✓ | ✓ | | | ✓ | | S | ✓ |
| Alizadeh et al. [64] | Pakistan | Stakeholder participatory approach used to study the coupled human–water system interaction under climate change scenarios, using SDM. | ✓ | ✓ | ✓ | ✓ | | ✓ | | | ✓ |
| Ghasemi et al. [65] | Iran | The water resource management, considering the regional socio-economic factors and stakeholder engagement. | ✓ | ✓ | ✓ | ✓ | ✓ | ✓ | ✓ | V | |
| Stave [66] | USA | The water management policy scenarios analyzed based on the stakeholder framework and SDM. | ✓ | ✓ | ✓ | | ✓ | ✓ | | V | |

### 4.5. Reuse of Treated Wastewater

Several scholars have emphasized the need to reuse treated wastewater (TWW) for effective water demand management. The urban WSD network of Tehran was modeled by Ghasemi et al. [65] using SDM to investigate the issue of the declining GW level. It was recommended that TWW should be reused not only for sanitary purposes but also for indirect GW aquifer recharge. Similarly, an assessment of water resource vulnerability in Chennai, India, was conducted by Rajarethinam et al. [67] using SDM. A water supply index and sewage index were developed for the system analysis. The recommended policies for improving the water system resilience included the development of sewage treatment plants and the reuse of treated sewage water for non-food irrigation. Xu et al. [68] studied the water resource management (WRM) of the Yellow River, China, analyzing the impact of TWW usage. GW is overexploited in the region, and 20% of the domestic water need is fulfilled using GW. An inter-basin water supply to the Yellow River, irrigation system efficiency, and the reuse of TWW were recommended. Similarly, WSDM in Bayinglin, China, was performed by Wu et al. considering TWW usage [69].

For environmental protection and to bring a balance to the WSD, a novel water return flow credit policy was developed by Qaiser et al. [70]. In this policy, the amount of water a commercial consumer can withdraw is equal to the amount of treated wastewater supplied to the system. The researchers found that TWW is essential for mitigating water resource challenges. The environmental flow (EF) of the Weihe Basin, China, was studied by Wei et al. [71] using SDM. EF is the essential water flow for the sustainability of the river ecosystem. For EF sustainability in the basin, an improved rate of TWW was found to be a significant factor.

The key findings of the section are tabulated in Table 5. The symbol (✓) means that the mentioned feature is considered in the research study.

### 4.6. Water–Energy–Food Nexus (WEFN)

An increasing food and energy demand leads to water security problems; this is called the WEF nexus [72], or the WEFN. Without a stable WEFN, sustainable development is difficult, especially for resource-based regions. The GW–energy–food (GEF) nexus concept was presented by Mirzaei et al. [73], revealing that the increasing food and energy demand poses a severe threat to GW levels, and reserves might face depletion in the near future. A DSS based on SDM was developed by Rahmani et al. [74] to evaluate GW management in the context of the WEF nexus in the Alboz Region, Iran. The declining level of the GW increases the energy demand for GW extraction. A combination of policies was advised,

including aquifer recharge, blocking illegal farm wells, irrigating low-water-consuming crops, and using solar-powered pumps for GW abstraction. In the Qazin region, Iran, WEFN policies were found to be interlinked as decreasing water levels increase the energy price for GW pumping, ultimately affecting food prices and availability [75].

**Table 5.** Reuse of treated wastewater reviewed studies. Reviewed articles on SD applications incorporating the reuse of treated wastewater. H: hydrological modeling; S: social modeling; Ec: economic modeling; En: environmental modeling.

| Reviewed Studies | Location | Modeling Approach | Sub-System Modeling | | | | SDM (System Dynamic Modeling) | | | Software Used | Climate Change Impact |
|---|---|---|---|---|---|---|---|---|---|---|---|
| | | | H | S | Ec | En | CLD | SFD | Validation | Vensim (V) or Stella (S) or Other (O) | |
| Ghasemi et al. [65] | Iran | GW sustainability analysis considering the reuse of treated wastewater, using SDM. | ✓ | | ✓ | ✓ | | ✓ | ✓ | V | |
| Rajarethinam et al. [67] | India | Water resource vulnerability assessment using SDM. | ✓ | ✓ | ✓ | | ✓ | ✓ | ✓ | S | |
| Xu et al. [68] | China | Water supply and demand assessment considering the impact of the reuse of treated wastewater and irrigation efficiency, using SDM. | ✓ | ✓ | ✓ | ✓ | ✓ | ✓ | | S | ✓ |
| Wu et al. [69] | China | Water resource vulnerability assessment using SDM. | ✓ | ✓ | | | ✓ | ✓ | | V | ✓ |
| Qaiser et al. [70] | USA | Water conservation assessment using SDM. | ✓ | ✓ | | | | ✓ | ✓ | S | |
| Wei et al. [71] | China | Assessment of environmental flow resulting from socio-economic scenarios, using SDM. | ✓ | ✓ | ✓ | | | ✓ | ✓ | V | ✓ |

The impact of anthropogenic activities on the WEFN in Daqing, China, was analyzed by Wen et al. [76]. The outcome showed that controlling anthropogenic activities to minimize water pollution and investment in water conservation development are essential for the future sustainability of natural resources and the associated socio-economic growth. Similarly, in Beijing, China, the WEFN was also analyzed by Li et al. [77]. A large gap between the WSD affecting the WEF was forecasted, with an increased dependence on imports if the baseline scenario prevails. To mitigate this challenge, a management plan known as the "Xiong Plan" was devised and suggested to policymakers. The SDM approach to the WEFN analysis proved a good tool for enhancing the system's resilience.

The key findings of the section are tabulated in Table 6. The symbol (✓) means that the mentioned feature is considered in the research study.

**Table 6.** Reviewed studies incorporating the water–energy–food nexus. Reviewed articles on SD applications in the context of the water–energy–food nexus. H: hydrological modeling; S: social modeling; Ec: economic modeling; En: environmental modeling.

| Reviewed Studies | Location | Modeling Approach | Sub-System Modeling | | | | SDM (System Dynamic Modeling) | | | Software Used | Climate Change Impact |
|---|---|---|---|---|---|---|---|---|---|---|---|
| | | | H | S | Ec | En | CLD | SFD | Validation | Vensim (V) or Stella (S) or Other (O) | |
| Rahmani et al. [74] | China | DSS development using SDM for GW resource management, using the WEF nexus approach. | ✓ | | ✓ | | | ✓ | ✓ | V | |
| Naderi et al. [75] | Iran | WEF approach towards water supply and demand management, using SDM. | ✓ | | ✓ | | ✓ | ✓ | | V | |
| Wen et al. [76] | China | SDM used to study the WEF nexus for sustainable socio-economic development. | ✓ | ✓ | ✓ | | ✓ | ✓ | ✓ | V | ✓ |
| Li et al. [77] | China | WEF resource security analysis using SDM. | ✓ | ✓ | ✓ | ✓ | | ✓ | ✓ | S | |

### 4.7. Water Quality Management

Climate variability, population growth, and increasing industrialization affect the WSD and negatively impact GW and surface water quality if not managed effectively [78]. Several anthropogenic effects directly impact water quality due to socio-economic growth, such

as nitrate contamination resulting from the excessive use of fertilizer and pesticides from agriculture or contaminant intrusion due to over-pumping and soil and water salinization [79]. Considering different socio-economic factors, several researchers have addressed the water quality management problem by using the SDM approach.

The water quality of the Mahabad reservoir, Iran, was studied by Nazari-Sharabianet al. [80]. Contamination due to total phosphorus (TP) was analyzed in different climate change and socio-economic scenarios. A hybrid research approach combining SWAT (soil and water assessment Tool) and SDM was used to investigate this problem. Under the several scenarios of climate change, population growth, industrial development, and agriculture and livestock production, an increase in TP loading was estimated with a maximum value of 108.9 µg/L, while an average 4% decrease in streamflow was estimated.

Considering socioeconomic development for the next 10 years, a strategic water management plan in China, the South–North Water Diversion (SNWD) project, was presented by Yang et al. [81]. The impact of the SNWD project on water quality and quantity was analyzed, considering the increasing population growth, expanding industries, and increasing agricultural production. Using SDM, it was found that the SNWD project will positively impact the water quality by reducing the pollution rate. For water conservation and environmental protection, the evaluation of managerial policies was carried out by Liu et al. [82] in the Dancha watershed, China. The water quality parameters analyzed included the COD (chemical oxygen demand), TN (total nitrogen), and TP (total phosphorous). Three management policy scenarios were designed and evaluated by simulation using SDM over a period of five years. It was concluded that developing a wastewater treatment plant will positively impact regional socio-economic development and water resource conservation. This study identified the significance of water resource preservation in terms of quality for long-term sustainable development.

The key findings of the section are tabulated in Table 7. The symbol (✓) means that the mentioned feature is considered in the research study.

**Table 7.** Water quality management reviewed studies. Reviewed articles on SD applications incorporating water quality management. H: hydrological modeling; S: social modeling; Ec: economic modeling; En: environmental modeling.

| Reviewed Studies | Location | Modeling Approach | Sub-System Modeling | | | | SDM (System Dynamic Modeling) | | | Software Used | Climate Change Impact |
|---|---|---|---|---|---|---|---|---|---|---|---|
| | | | H | S | Ec | En | CLD | SFD | Validation | Vensim (V) or Stella (S) or Other (O) | |
| El Mountassir et al. [79] | Sweden | Sustainability assessment of a contamination site using SDM to improve the resilience of water quality and pollution management system. | | ✓ | ✓ | ✓ | ✓ | ✓ | | S | |
| Nazari-Sharabian et al. [80] | Iran | Water quality management using SDM and SWAT approach. | ✓ | ✓ | ✓ | ✓ | | ✓ | ✓ | S | ✓ |
| Yang et al. [81] | China | Water quality and quantity management using the SDM approach for a SNWD strategic water plan. | ✓ | | ✓ | ✓ | ✓ | | | V | |
| Liu et al. [82] | Iran | SDM for water quality management, considering the impact of waste-water treatment plant. | ✓ | ✓ | ✓ | ✓ | ✓ | ✓ | ✓ | V | |

## 5. Quantitative Techniques and Hybrid Models

This section discusses quantitative techniques, including optimization and analytical methods that address water resource allocation optimization. Water allocation problems have been formulated and actively solved using optimization techniques [83].

### 5.1. Optimization Techniques

To guide policymakers in making better decisions, a hybrid optimized SDM approach was used. Goorani and Shabanlou [84] used this approach to improve the quality and quantity of a water resource system in a wetland in Shedgan, Iran. With an imbalance in upstream water resources, agriculture drainage, and municipal wastewater discharge,

the wetland faced a reduced environmental flow with increased salinity. An optimization model based on the NSGA-II algorithm was developed to address this challenge. The parameters were simulated using a WEAP (water evaluation and planning) analytical model. The system reliability was improved by 85% in terms of efficient water availability for the wetland, and the salinity of the inflow was reduced by 40%. Similarly, the water supply for hydropower generation in Karun, Iran, was optimized using a simulation optimization method developed by Majedi et al. [85]. The water balance was achieved using the MODFLOW and WEAP models, while the NSGA-II optimization method was used to minimize the demand shortages and the aquifer abstraction rate. Safavi et al. [86] optimized a management policy for the conjunctive use of ground and surface water in a semi-arid basin of Iran. A simulation optimization technique was used to simulate the interaction between groundwater and surface water using ANNs, and it was solved using a genetic algorithm (GA). The developed model aided the decision makers in effectively managing the conjunctive use of GW and surface water in semi-arid regions.

Optimization approaches are best-suited to economic modeling. To boost economic growth and improve social indicators (increased job opportunities), a bi-objective optimization model was employed by Habibi Davijani et al. [87]. The model was applied as a case study using optimization modeling in an arid region of Iran, with the prime objective of improving the socio-economic indicators by efficiently allocating water. Two meta-heuristic methods, GA and particle swarm optimization (PSO), were applied to solve the model. The PSO solution promised better results, with an overall increase of 40% in employment rates. Using a hybrid simulation optimization approach, the water management and allocation strategies for the Najaf Basin, Iran, were analyzed by Naghdi et al. [88]. The region faces a water security problem due to the increasing water demand and the overexploitation of GW resources. GW overexploitation is practiced to fulfil irrigation water demands. For the efficient allocation of groundwater and surface water among users, the NSGA-II optimization technique was used. WSD optimization in Singapore was performed by Xu et al. [89] using bi-objective optimization and SDM. Water production, distribution, and treatment cost were modeled and minimized. To aid decision makers in policy-making scenarios, the researchers suggested a water management plan for the future 15 years, using this novel optimization simulation model. Simultaneous socio-economic and environmental optimization is necessary for sustainable development. The authors of [90] created a framework that includes prediction, optimization, and decision-making models that employ multi-objective optimization to distribute the available water resources among the various sectors efficiently.

The key findings of the section are tabulated in Table 8. The symbol (✓) means that the mentioned feature is considered in the research study.

The literature has revealed that water allocation problems are addressed to optimize water WSDM using multi-objective optimization techniques, but little research has been found on hybrid SDM and optimization methods to solve water resource allocation problems [91–93].

### 5.2. Hybrid SD Models

Researchers have analyzed several water management and planning strategies by combining SDM with other analytical methods to further improve system understanding [94]. The DPSIR (driver-pressure-state-impact-response) method has been used to clearly scope problem scoping and understanding. The DPSIR and SDM were merged by Zare et al. [95] as the hybrid DPSIR-SDM approach to address the water resource management of Gurganrud Basin, Iran. Ahmad and Al-Ghouti [96] studied GW sustainability in Qatar using the DPSIR method. The GW aquifer in Qatar is under high stress due to over-exploitation, mainly for agricultural needs. The impact is a declining GW table and increased salinity due to seawater intrusion. It was recommended that farmer awareness campaigns be initiated to encourage farmers to use treated sewage effluent (TSE) for irrigation. The artificial recharge of aquifers using TSE was also advised.

**Table 8.** Optimization techniques in the reviewed water resource management articles. Reviewed articles on SD applications complemented by optimization techniques. H: hydrological modeling; S: social modeling; Ec: economic modeling; En: environmental modeling.

| Reviewed Studies | Location | Modeling Approach | Sub-System Modeling | | | | SDM (System Dynamic Modeling) | | | Simulation Model | Optimization | Climate Change Impact |
| | | | H | S | Ec | En | CLD | SFD | Validation | Vensim (V) or Stella (S) or Other (O) | Solution Approach | |
|---|---|---|---|---|---|---|---|---|---|---|---|---|
| Goorani, and Shabanlou [84] | Iran | Water demand and quality management using WEAP (Simulation) and NSGA-II (multi-objective optimization). | ✓ | | | ✓ | | | | O (WEAP) | Multi-Objective-NSGA-II | |
| Majedi et al. [85] | Iran | Water resource management using MODFLOW, WEAP (Simulation), and NSGA-II (multi-objective optimization). | ✓ | | | ✓ | | | ✓ | O (MOD-FLOW and WEAP) | Multi-Objective-NSGA-II | |
| Safavi et al. [86] | Iran | Simulation optimization approach to managing the conjunctive use of surface water and GW. | ✓ | | ✓ | ✓ | | | ✓ | O (ANN) | GA | |
| Habibi Davijani et al. [87] | Iran | Water allocation optimization using a bi-objective socio-economic model to improve the socio-economic indicators (employment rate). | ✓ | ✓ | ✓ | | | | | | Bi-objective model GA, PSO | |
| Naghdi et al. [88] | Iran | Water allocation problem using the SDM (simulation), NSGA-II (optimization), and Nash bargain method (conflict resolution). | ✓ | ✓ | | ✓ | ✓ | ✓ | ✓ | | Multi-Objective-NSGA-II | ✓ |
| Xu et al. [89] | Singapore | Water allocation optimisation using a bi-objective model (optimization) and using the SDM (simulation) for scenario analysis. | ✓ | | ✓ | ✓ | ✓ | ✓ | ✓ | V | Bi-objective model (LINGO 20-software) | |

To diversify and increase the water supply of Singapore, several water infrastructure investment options were analyzed and evaluated using a novel approach: the integration of SDM with the AHP by Xi and Poh [97]. The WSD of the region was simulated using SDM with three population growth scenarios. Different water infrastructure development plans were embedded in the simulation model to analyze the impact on the water supply. The water infrastructure development plans were prioritized and ranked using the AHP methodology. It was found that GW storage development is not a viable long-term solution, considering the population growth, while the development of desalination plants serves the purpose well and helps in the sustainable socio-economic development of Singapore.

It is also essential to have a comprehensive socio-economic analysis of bio-physical water resource systems. This requires coupling physical-process-based software (e.g., MODFLOW and SAHYSMOD) that analyzes the GW flow and soil salinity and SDM software (e.g., VENSIM 9.3.5, STELLA 2.0, etc.). Malard et al. [98] developed an open-source physical-process-based software in Python called Tinamit, which bridges this gap. The soil salinity of a selected region of Pakistan was analyzed.

A hybrid SDM-TOPSIS (technique for order of preference by similarity to ideal solutions) was used by Rezaee et al. [99] to evaluate the water system of the Azerbaijan region of Iran. Thirteen policy scenarios were simulated, and it was found that the current practice scenario ranked sixth, indicating the need for improvement. The suggested methodology can be used by policymakers to efficiently rank different water demand sectors and allocate water according to the priority of their identified ranking.

Using an expert system based on the adaptive-network-based fuzzy inference system (ANFIS), an integrated water resource study of the Zayandeh Basin, Iran, was conducted by Safavi et al. [100]. The basin condition in the coming five years was projected, considering the current water management strategies and climate conditions. The overexploitation of GW must be controlled through effective management practices.

A comparative analysis of water resource management policies in the Kairouan region, Tunisia, was conducted by Sušnik et al. [101]. SDM and an object-oriented Bayesian network (OOBN) were used to analyze the policy's impact on water resources. SDM captured the dynamic, non-linear behavior of the system, whereas the OOBN captured the uncertainty in terms of the input and output data. A sustainability analysis of seven coastal lakes in Australia was conducted by Croke et al. [102] using Bayesian network modeling (BNM). A participatory workshop was arranged to gather stakeholder input to understand the system. Several management policy scenarios were devised, and using the stakeholder input, the probable implications on lake health were evaluated to minimize the negative impacts. The effect of the behavior of different agents on the GW resources of the

Zayndeh Basin, Iran, was analyzed by Ohab-Yazdi and Ahmadi [103] using the agent-based simulation model (ABM). All the regional stakeholders were identified, and the Regional Government Water Department (RGWD) was found to be the most significant agent.

The effect of the dynamic interaction of agents and stakeholders on GW resources was modeled and analyzed using SDM in the AnyLogic 8.8.2 software. The results revealed that the stakeholders' effective interaction with RGWD reduced illegal GW abstraction and improved water sustainability. Sustainable water allocation and demand management strategies were analyzed by Arasteh and Farjami [104] in Yazd, Iran. A combination of SDM and agent-based modeling (ABM) was used to analyze eighteen water management policy scenarios. The combination of a smart water metering system, a medium increase in water price, and the reuse of treated wastewater and grey water for irrigation practices were recommended.

The key findings of the section are tabulated in Table 9. The symbol (✓) means that the mentioned feature is considered in the research study.

**Table 9.** Hybrid SD reviewed studies. Reviewed articles on hybrid SD studies. H: hydrological modeling; S: social modeling; Ec: economic modeling; En: environmental modeling.

| Reviewed Studies | Location | Modeling Approach | Sub-System Modeling | | | | SDM (System Dynamic Modeling) | | | Method Used | |
| | | | H | S | Ec | En | CLD | SFD | Validation | Technique | Climate Change Impact |
|---|---|---|---|---|---|---|---|---|---|---|---|
| Zare et al. [95] | Iran | Water resource management using hybrid SD-DPSIR modeling approach. | ✓ | ✓ | ✓ | | ✓ | ✓ | ✓ | DPSIR | |
| Ahmad, and Al-Ghouti [96] | Qatar | GW sustainability assessment using DPSIR methodology. | ✓ | ✓ | ✓ | | | | | DPSIR | ✓ |
| Xi and Poh [97] | Singapore | Water resource management using hybrid SD-AHP modeling. | ✓ | | ✓ | | ✓ | ✓ | ✓ | AHP | |
| Malard et al. [98] | Pakistan | GW resource assessment using an open-source DSS Tinamit employing the SD approach. | ✓ | ✓ | ✓ | ✓ | ✓ | | | Tinamit DSS | |
| Rezaee et al. [99] | Iran | Water resource management using hybrid SD-TOPSIS framework. | ✓ | ✓ | ✓ | ✓ | | ✓ | | TOPSIS | |
| Safavi et al. [100] | Iran | Water resource management using an expert-system-based ANFIS modeling approach. | ✓ | ✓ | | | | | | ANFIS | |
| Sušnik et al. [101] | Tunisia | Water resource management using hybrid SD-BN modeling approach. | ✓ | | ✓ | ✓ | ✓ | | | Bayesian Network | ✓ |
| Croke et al. [102] | Australia | Water resource management ensuring stakeholder participation, using BN modeling. | ✓ | ✓ | | | | | | Bayesian Network | ✓ |
| Ohab-Yazdi and Ahmadi [103] | Iran | GW sustainability assessment using ABM and SDM. | ✓ | ✓ | ✓ | | | | | Agent Based Modeling (AnyLogic Simulation Software) | |
| Arasteh and Farjami [104] | Iran | Water demand management considering socio-hydro sub-systems, using ABM and SDM. | ✓ | ✓ | ✓ | | ✓ | ✓ | ✓ | Agent Based Modeling | ✓ |

## 6. Discussion

This study critically analyzed the application of SDM to sustainable WSDM through the systematic review of sixty-nine research articles. WSDM using SDM was applied to study regional sustainable development, effective irrigation management, the balancing of the water–energy–food nexus, the mitigation of the impact of climate change, the control of GW overexploitation, water quality management, and the fruitful engagement of stakeholders to identify and implement pragmatic water resource management solutions.

A total of 88% of the reviewed SDM applications employed a scenario-based analysis approach to understand the current dynamic system, evaluate different possible solution scenarios, and test the effectiveness of the adopted water management policies and strategies, as shown in Tables 1–9. Hybrid SDM approaches using the multi-criteria decision modeling (OB, ABM, and simulation optimization modeling approaches) were employed by only 12% of the reviewed studies, as shown in Tables 8 and 9. This may be attributed to

the need for an interdisciplinary research team to conceptualize and model water planning and management strategies using complex hybrid approaches, as Figure 3 shows.

The conceptualization stage of SDM using CLD is an essential qualitative tool for problem framing and scoping, yet only 58% of the reviewed studies developed CLD in their models. A quantitative analysis using SFD was only performed by 70% of the reviewed studies. Model validation is essential for comparing the simulated model with the actual system and is necessary to ensure that the simulated model replicates the actual system's attributes of interest under investigation; this was performed by 59% of the reviewed studies. The impact of climate change on water resource management is one of the essential sub-categories to be analyzed, especially in arid or semi-arid regions. Only 51% of the studies incorporated the impact of climate change into SDM to analyze the water resource management strategies. VENSIM was the most widely used simulation software for SDM, being used by 47%, while STELLA was used by 23%, as shown in Tables 1–9.

Strategic policies play a pivotal role in the long-term sustainable development of any region—88% of the reviewed articles performed a scenario-based policy evaluation using the SDM approach. Socio-economic-environmental indicators must be incorporated into the model for effective policy making. Consequently, the recommended SDM is the most suitable approach [8], which uses a holistic system approach to address WSDM challenges. Researchers suggested a couple of policies to contribute to regional sustainable development, including the regulation of the per capita water consumption rate [15] and the development of water infrastructure projects that consider future water demand [26] and inter-basin water diversion and transfer [27]. The balance between water supply and demand due to increasing energy needs revealed that energy demands significantly affect water management. SDM helps policymakers to understand and holistically analyze the system [30]. The previously mentioned study supports research questions 1 and 2, as SDM is an effective decision support tool for WSDM, fostering sustainable development.

Furthermore, SDM aids decision makers in devising strategies that foster GW conservation, supporting question 3. Using the SDM approach, ref. [37] employed sustainability indicators, including GW equity, sustainability, and consumption efficiency, to evaluate the smartness of GW governance policies. The authors of [38] demonstrated that by using SDM, GW overexploitation can be controlled by reducing the water demand per capita, especially in coastal and arid regions. Furthermore, the positive impact of water conservation policies, considering modern irrigation techniques, including drip irrigation, the cultivation of low-water-consuming crops, and scenarios of different water pricing schemes, was simulated and analyzed using SDM [45,48,50,60]. Specifically, the impact of artificial aquifer recharge using TSE was simulated by the SDM approach and was found to be a fruitful technique for GW conservation [67]. Consequently, policymakers can use SDM to analyze the long-term impact of policies on water conservation.

To improve the accuracy of system representation, hybrid SDM models incorporate components from two or more different types of modeling methodologies, including ABM, MCDM, and optimization techniques. These models can be especially helpful when dealing with complicated systems with several interrelated components. To complement the SDM approach, ref. [95] used the DPSIR method to better conceptualize water resource challenges. Using the same approach, GW stress indicators were identified [96]. Using AHP with SDM, several water infrastructure projects were evaluated and ranked [97]. A bi-objective optimization was developed by [87] to model WSDM. GA and PSO heuristics were used to solve the model. Water demands were satisfied using a hybrid optimized SDM framework while improving the water quality by reducing water salinity [84]. A hybrid SDM-TOPSIS technique assessed and ranked water management policies [99]. Effective water demand management strategies were devised and implemented using a hybrid SDM-ABM approach [104]. These studies addressed research question 4, "What are the different quantitative models used in conjunction with SDM to improve water resource management?".

One of the drawbacks of applying optimization approaches alone to water resource management problems is that social constraints are difficult to capture in a quantitative

model. Thus, the solutions proposed by the optimization models often lead to lower acceptability [88]. Therefore, to effectively address WSDM challenges, hybrid techniques that integrate SDM, ensuring stakeholder engagement, are recommended.

## 7. Conclusions

In summary, SDM is useful for managing sustainable water supply and demand, especially when incorporating stakeholders' input into the model. The complex dynamic WSDM, considering socio-economic and environmental sub-systems, can be effectively conceptualized and analyzed quantitatively using CLD and SFD simulation. The non-linear, dynamic simulation of a complex water system makes it feasible for decision makers to devise and test the long-term impact of water policies on the overall system. Hence, SDM is an effective tool for sustainable WSDM and sustainable societal development.

It was noted that half of the studies did not incorporate the impact of climate change into their models, which makes the studies less effective, as the changing climate significantly impacts the surface water and GW resources. Hence, the effects of climate change on water resource management need to be considered in future studies.

Finally, it was found that although several studies employed pure optimization approaches to develop a single or bi-objective model to solve water supply and demand problems, only a few studies incorporated SDM to complement their optimization models. Specifically, most researchers focused on surface water or TSE, and few addressed the GW conservation objective.

Several management strategies have been recommended to improve the water resource management system, including a reduction in the per capita water consumption, the use of treated wastewater, investment in wastewater treatment plants, regulations to control the GW abstractions, public awareness campaigns to instill water conservation strategies, and the improvement of irrigation management practices, including the cultivation of low-water-consuming crops.

## 8. Future Recommendation

For comprehensive water supply and demand management, all water sources must be considered to aim for water conversation. SDM helps to clarify system understanding and identify and model social issues in water resource management, which are difficult to capture in optimization models. The limitation of SDM is that it is a simulation scenario-based approach and cannot guarantee optimum results. Hence, in future research, developing a simulation optimization model that employs SDM is suggested to help policymakers to clearly understand the interaction of dynamic socio-economic factors, optimize system parameters, and devise pragmatic strategies that are socially accepted. Specifically, an optimized SDM approach that considers multiple objectives is advised. Water demand fulfillment, GW conservation, and environmental sustainability are the critical objectives to be optimized. For the solution approach, NSGA-II is recommended. Strategic decision making using the developed research will help in sustainable socio-economic development.

Additionally, a limited number of authors focused on solving the WEF nexus challenges using a simulation optimization approach, especially in arid regions. This could be a pragmatic future research direction to foster regional sustainable development.

Analyzing water demand, most of the research articles have focused on the domestic, industrial, agricultural, and environmental sub-sectors. None of the research articles focused on the sharp increase in water demand due to mega events in the region for a short period. This could be another interest research gap to be addressed.

Furthermore, no research was found that explored the impact of water conservation using hydroponic farming for agriculture instead of traditional agriculture practices. This could also be another future research direction that would alleviate the food security challenge.

**Author Contributions:** All authors collaborated in the research presented in this publication by making the following contributions: research conceptualization, K.N., A.Z., A.E., A.M. and C.T.; methodology, K.N., A.Z., A.E., A.M. and C.T.; formal analysis, K.N., A.Z., A.E., A.M. and C.T.; writing—original draft preparation, K.N. and A.Z.; writing—review and editing, K.N., A.Z., A.E., A.M. and C.T.; supervision, A.E., A.M. and C.T. All authors have read and agreed to the published version of the manuscript.

**Funding:** This publication was made possible by BFC grant #BFC04-0719-200004 from the Qatar National Research Fund (a member of Qatar foundation). The findings herein reflect the work, and are solely the responsibility of the authors.

**Data Availability Statement:** All data are contained in the article.

**Conflicts of Interest:** The authors declare no conflict of interest.

## Abbreviations

| | |
|---|---|
| ABM | Agent-Based Modeling |
| AHP | Analytical Hierarchy Approach |
| ANN | Artificial Neural Networks |
| AOGCM | Atmosphere–Ocean General Circulation Model |
| BN | Bayesian Networking |
| BNM | Bayesian Network Modeling |
| CLD | Causal-Loop-Diagram |
| COD | Chemical Oxygen Demand |
| DPSIR | Driver-Pressure-State-Impact-Response |
| DSS | Decision Support System |
| DSS-ESPA | Decision Support System–Eastern Plain Aquifer |
| DW | Desalinated Water |
| EF | Environmental Flow |
| GA | Genetic Algorithm |
| GHG | Greenhouse Gases |
| GMS | GW Management System |
| GW | Groundwater |
| MOO | Multi-Objective Optimization |
| NBS | Nature-Based Solutions |
| NSGA | Non-dominated Sorting Genetic Algorithm. |
| OOBN | Object-Oriented Bayesian Network |
| PSO | Particle-Swarm-Optimization |
| SDM | System Dynamics Model |
| SEHEc | Social, environment, hydrological, and economic |
| SFD | Stock-Flow-Diagram |
| SNWD | South–North Water Diversion |
| SP | Stakeholder Participation |
| SW | Surface Water |
| SWAT | Soil and Water Assessment Tool |
| TN | Total Nitrogen |
| TOPSIS | Technique for Order of Preference by Similarity to Ideal Solutions |
| TP | Total Phosphorous |
| TSE | Treated Sewage Effluent |
| WEAP | Water Evaluation and Planning |
| WEFN | Water Energy Food Nexus |
| WRM | Water resource management |
| WSD | Water supply and demand |
| WSDM | Water supply and demand management |

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
