# Peer review of "A Literature Review on System Dynamics Modeling for Sustainable Management of Water Supply and Demand"

_sustainability, doi:10.3390/su15086826_

Round 1
Reviewer 1 Report
Review report - Sustainability 2253631 – Manuscript
Summary of article
Aim of the paper:
· This research presents a thorough analysis of the use of SDM to sustainable WSDM. Over sixtieth-nine studies published within the last two decades were evaluated.
· Agent-Based Modeling (ABM), Bayesian Networking (BN), Analytical Hierarchy Approach (AHP), simulation-optimization M-Objective Optimization (MOO), and Genetic Algorithm (GA), Particle-Swarm-Optimization (PSO), and Non-dominated Sorting Genetic Algorithm (NSGA) are quantitative models that complement the SDM.
Main contributions:
The following main contributions were observed with supporting results
· The key findings for water resource sustainable development included per capita water reduction, water conservation through public awareness campaigns, the use of treated wastewater,
· The adoption of efficient irrigation practices including drip irrigation, the cultivation of low-water consuming crops in water-stressed regions, and regulations to control groundwater overexploitation.
· For future research, a hybrid SDM-Optimization framework is proposed to simulate and optimize the dynamic socio-economic interactions and assist policymakers in formulating strategic water planning and management policies that promote sustainable development.
Strengths of article: Study methodology is designed in an elaborate and critical manner.
Comments on article
· Article is needs major improvements throughout manuscript.
· Figures and Tables need to represented in a focused manner for clarity to readers.
· Appropriate Referencing needs to be improved in entire manuscript.
· Latest references need to be included in manuscript.
Specific comments
Abstract section
· Abstract is not conveying the information of study clearly. Details of aims, methodology and results observed are clearly represented.
· Kindly rewrite entire abstract. Include Objectives, methodology and results observed in a concise manner.
Introduction section
· Introduction is not focused on the research topic of article and contains theoretical information.
· Combine section 1 and 2 into introduction section.
· Figure 1: focus the figure on the design and concept of the research study. Theoretical and academic context information about SDM may not be interest to readers.
· Over all, Introduction needs to be rewritten focusing on the research concept of article.
Methodology
· Figure 2: Put x and y axis values, measurements of the bar graph.
· Figure 2: Try to redesign the figure focusing on various factors and variables of study design in past 5 year’s relevance.20 years back information about research articles, may not be supportive to the latest developments in relevance to current topic of research.
· Figure 3: Try to improve the research design in a more critical manner. Do not repeat already mentioned information in text.
· Figure 4: Put x and y axis values, measurements of the bar graph. Reconsider, if this figure is supporting to provide insightful information towards current research design.
· Figure 5: This figure is not necessary and is not conveying any critical information.
· Figure 6: Improve the figure representation and make figure more insightful by providing critical factors relating to study design.
Results
· Rename the section as results by replacing the subheading content analysis
· Table 1: Table is not conveying any useful information to readers, by putting the reference no’s of articles.
· Table 2: Improve table representation
· Table 3: Improve table representation
· Table 4: Improve table representation
· Table 5: Improve table representation
· Table 6: Improve table representation
· Table 7: Improve table representation
· Table 8: Improve table representation
· Table 9: Improve table representation
· Table 10: Improve table representation
· Figure 7: Put x and y axis values, measurements of the bar graph.
· Too many tables, try to represent critical findings in 4 to 5 tables for more clarity to readers.
Discussion
· Improve discussion section by citing relevant figures and tables
Conclusions
· Rewrite conclusion clearly summarizing the main observations of study.
· Include a section on future recommendations for this research topic.
References
· Most of references included in study are quite old.
· Please include information from latest reference articles as much as possible.

Author Response
The authors would like to thank the reviewer for their valuable comments. We took into account almost all comments. They are introduced in the modified text highlighted with Word Track Changes.

Reviewer 2 Report
This article is very usefull to the researches, but it needs a major review. Please see the attached comments below.

Author Response

(The authors gave the same response as above.)

Reviewer 3 Report
Dear Authors
The following modifications are required:
Title
Point 1. The title should be changed to:
An Overview of System Dynamics Modeling as a Tool for Sustainable Water Supply and Demand Management
Abstract
Point 2. L17 (This paper provides a comprehensive review of the application of SDM to 17 sustainable WSDM): The authors should detail the aim of this review because it is simply written. Point 3. The authors should explain the advantage of using SDM.
Point 4. L18 (Over sixty-nine papers spanning over the last 20 years were analyzed): This sentence has no sense here and should be deleted
Point 5. L25: A brief conclusion about the review article should be added
Keywords
Point 6. The terms used in the forming of the title should not be used as the keywords, so the content of keywords should be changed
Introduction
Point 7. L34: The authors should provide some information about the term: water supply system
Point 8. L41-44 (The climate change impact further exacerbates the problem, especially in arid and semi-arid regions where groundwater (GW) is the sole natural water resource and is over exploited due to poor long-term water management strategies [5]. This leads to the problem of GW depletion, seawater intrusion and the degradation of the ecosystem [6]): These sentences should be placed at the beginning of the introduction.
Point 9. In the introduction section, some lines about the System Dynamics Modeling and its levels of types should provide.
Point 10. L45-53 (The issue of GWM is further muddled because of the lack of updated and uncertain data, making analysis difficult [8]. The problem is further exacerbated by the conflicting objectives of multiple stakeholders and governing institutes): These lines should be deleted
Point 11. The authors should provide some information computer technology advances and mathematical programming that used in the study of System Dynamics Modeling.
Point 12. The captions of all figures are simply written and should be detailed. The title should reflect the content of figures
Point 13. L222-225 (The long-term impact of five different water policies is studied to mitigate the water shortage problem. The authors recommended conducting the inter-basin water transfer and improve the waste-water treatment efficiency. This will reduce the water shortage by approximately 40% in the coming 10 years): These lines have no sense and should be deleted
Point 14. L226-230 9 Shenzhen, China, faced a water scarcity challenges due to the expansion of the industries in the region. To address this challenge, the city's water distribution system is simulated for a fifteen-year time span using the SD approach by Li et al. [30]. It is found that to maintain and improve the water balance, wastewater reuse and improved water use efficiency both from domestic and commercial consumers are recommended): These lines have no sense and should be modified
Point 15. L271-273 (Huang et al. [45] analyzed the land subsidence in Taiwan using SDM, which is caused due to the over-exploitation of GW. Sealing the unauthorized wells and controlling the illegal GW extraction in the region is recommended): This line should remove
Point 16. L278-281(The interaction between GW-surface water in China is studied by Khan et al. [48] using the SDM. The research objective is to enhance water use efficiency under different irrigation scenarios. This assisted the decision-makers in devising strategies that support sustainable irrigation development): These lines should be improve because they have no sense
Point 17. The caption of all Tables should be detailed and reflected the content of the table.
Point 18. The review of all articles is not well presented and should be improved
Point 19. Most of the articles used in this review are originated from the Asia continent and the authors should expand the review content by using the review articles originated from different continents
Point 20. The discussion is too weak and should be improved
Author Response

(The authors gave the same response as above.)

Reviewer 4 Report
Sustainability-2253631 provides some valuable information to the researchers and readers. The subject of the manuscript is consistent with the scope of the Journal. I suggested that the manuscript need to be major revised before it is accepted by this journal.
1. To write a good review, one needs to have a clear aim, consequently a review should start with an overview of what reviews exist already on the topic and how your review compares to these, e.g. is it an update, or an analysis of the existing knowledge from a different point of view.
2. Abstract and conclusion are very generic and should summarise the purpose and what is presented as new within the review. Future perspectives are important but too importance is given to them. I suggest to focus on current trends.
3. A review should be more than a compilation of the results reported in literature (i.e. just copy and paste), it should in fact be a critical assessment of the present knowledge with some clear conclusions what all these results mean, and what are the perspectives and directions for future research and potential applications. This is not clear within review, please adjust to my suggestion.
4. A list of abbreviations should be provided for reading.
5. It is suggested that the manuscript needs language editing. The research described in this manuscript is made more difficult to understand technically by the difficult-to-understand English language usage. It is noted that your manuscript needs careful editing by someone with expertise in technical English editing paying particular attention to English grammar, spelling, and sentence structure so that the goals and results of the study are clear to the reader.
6. The logic and neat of introduction need to be further improved, too many paragraphs seem disorderly.
7. The neatness of the article must be improved. There are too many paragraphs, some of which are long and some of which are short, which appear to be disorderly.
8. Conclusion should be further improved and shortened.
Author Response

(The authors gave the same response as above.)

Round 2
Reviewer 1 Report
Comments
Arrange figures and tables in a easily readable format.
Improve representation of figures and tables for easy understanding of readers.
Reviewer 2 Report
The article is now ready to be published.
Reviewer 3 Report
The authors have been addressed all comments
Reviewer 4 Report
I have no comment.